# One-Pot Hydrothermal Preparation of Fe_3_O_4_ Decorated Graphene for Microwave Absorption

**DOI:** 10.3390/ma13143065

**Published:** 2020-07-09

**Authors:** Zhonghe Du, Xibang Chen, Youwei Zhang, Xueyan Que, Pinggui Liu, Xiuqin Zhang, Hui-Ling Ma, Maolin Zhai

**Affiliations:** 1Beijing Key Laboratory of Clothing Materials R & D and Assessment, Beijing Engineering Research Center of Textile Nanofiber, School of Materials Science & Engineering, Beijing Institute of Fashion Technology, Beijing 100029, China; 18332762789@163.com (Z.D.); clyzxq@bift.edu.cn (X.Z.); 2Beijing National Laboratory for Molecular Sciences, Department of Applied Chemistry and the Key Laboratory of Polymer Chemistry and Physics of the Ministry of Education, College of Chemistry and Molecular Engineering, Peking University, Beijing 100871, China; xbchen@pku.edu.cn (X.C.); xy_que@pku.edu.cn (X.Q.); 3Beijing Institute of Aeronautical Materials, Beijing 100095, China; ywzhang_pku@163.com (Y.Z.); liupinggui@139.com (P.L.)

**Keywords:** microwave absorption, hydrothermal method, graphene, Fe_3_O_4_ nanoparticles, nanocomposites

## Abstract

Fe_3_O_4_ decorated graphene was synthesized for electromagnetic wave absorption via a facile one-pot hydrothermal approach. The structure and morphology of the as-prepared nanomaterials were systematically investigated. The graphene oxide (GO) was reduced and Fe_3_O_4_ nanoparticles were evenly decorated on the surface of reduced graphene oxide (rGO) nanosheets. The average particle size of Fe_3_O_4_ nanoparticles is about 15.3 nm. The as-prepared rGO-Fe_3_O_4_ nanocomposites exhibited a good microwave absorption performance because of the combination of graphene and magnetic Fe_3_O_4_. When the thicknesses are 1.6 mm and 6.5 mm, the reflection loss (RL) values are up to −34.4 dB and −37.5 dB, respectively. The effective bandwidths are 3.8 and 1.9 GHz.

## 1. Introduction

With the extensive usage of digital devices, the electromagnetic wave (EMW) is becoming a new environmental pollutant and threat. It not only seriously disrupts the operation of electronic equipment and satellite communication but also jeopardizes human health [1,2]. EMW absorbing materials are considered to be the most effective strategy to eliminate EMW pollution because they can absorb EMW and convert the electromagnetic energy into heat loss. Therefore, there is an urgent demand for developing high-performance EMW absorbing materials to counteract the adverse effects [3].

The ideal EMW absorbing materials should possess strong absorption ability, broadband, and low density [4]. To serve these purposes, various materials have been investigated, such as magnetic metal powders [5], ferrite [6], and titanates [7]. However, these materials cannot meet the increasing requirements of the EMW absorbing materials due to drawbacks like high density and being easily corroded or oxidized in harsh environments. To overcome these problems, carbon-based materials including graphite [8,9] and carbon nanotubes [10,11] have been developed to improve the EMW absorption because of low density, good thermostability, and corrosion resistance. As a representative of novel carbon-based nanomaterials, graphene is considered as an excellent EMW absorbing material due to its prominent carrier mobility, high surface area, ultra-light weight, and high thermal and chemical stability [12]. However, the EMW absorption of pristine graphene is very weak because of its poor impedance matching caused by low permeability and high permittivity [13].

Graphene oxide (GO), as an important graphene precursor, exhibits some similar properties to graphene. It is widely regarded as a simple method to obtain graphene by reducing GO. These structural defects arising from oxygen-containing groups (hydroxyl, epoxy, and carbonyl) are facilitated to the multiple dielectric loss [14]. However, the EMW absorption property needs to be further improved because of the weak impedance matching. The most conventional approach to solve the problem is introducing magnetic nanoparticles on reduced graphene oxide (rGO). The introduced magnetic materials are mainly metallic magnetic nanoparticles, such as Co, Fe, and Ni [15,16,17]. Ding et al. synthesized a CoFe@rGO nanocomposite via a three-step chemical method by using the raw materials of FeCl_3_·6H_2_O, CoCl_2_·6H_2_O and rGO. The CoFe@rGO nanocomposites exhibited a maximum reflection loss of −25.66 dB at 16.63 GHz [18]. However, these metallic nanoparticles with high electrical conductivities may decrease magnetization because of the eddy current effect [19]. Furthermore, these nanoparticles are easily oxidized and corroded. Compared with metallic nanoparticles, Fe_3_O_4_ nanoparticles are commonly considered as an important candidate for EMW absorbing materials because it possesses low electrical conductivity, good corrosion resistance and oxidation resistance [20]. Herein, we are motivated to prepare Fe_3_O_4_ nanoparticle decorated rGO nanocomposites (rGO–Fe_3_O_4_) via a simple hydrothermal treatment. The structure and morphology, EMW absorption properties in high frequency and low frequency, and the mechanism of the obtained products are systematically investigated.

## 2. Experimental

### 2.1. Materials

GO powders were bought from the Sixth Element Materials Technology company (Jiangsu, China). The FeCl_3_·6H_2_O, polyethylene glycol (PEG) and ethylene glycol were supplied by the Xilong Chemical company (Beijing, China). Sodium acetate (CH_3_COONa) was purchased from the Tianjin Huadong Reagent Factory (Tianjin, China).

### 2.2. Preparation of rGO-Fe_3_O_4_

GO powders (100 mg) were added to ethylene glycol (100 mL) and ultrasonicated for 1.5 h. Concurrently, 2.4 g FeCl_3_·6H_2_O, 2.4 g CH_3_COONa, and 5.4 g PEG were added into 30 mL ethylene glycol. The two solutions were transferred into a 150 mL Teflon-lined stainless steel autoclave. After reacting at 200 °C for 12 h, the product (rGO-Fe_3_O_4_) was separated by magnet and washed using distilled water and ethanol, then dried by freeze drying for 1 day. The bare Fe_3_O_4_ and rGO were also prepared in a similar process except for the addition of GO or FeCl_3_·6H_2_O, respectively.

### 2.3. Characterization

The products were characterized through X-ray diffraction (XRD, Rigaku D/Max 2400 diffractometer, Tokyo, Japan) with Cu Kα source at a scanning speed of 5°/min from 10 to 70°. GO, rGO, Fe_3_O_4_, and rGO-Fe_3_O_4_ were analyzed with a Fourier transform infrared spectrometer (FTIR, Thermo Fisher Scientific Nicolet iS 50, Waltham, MA, USA). Raman spectra (Thermo Scientific DXRxi, Waltham, MA, USA) were recorded by a Raman system with an excitation wavelength of 514.5 nm. Thermogravimetric analysis (TGA, PerkinElmer TGA 8000, Waltham, MA, USA) was carried out using a thermal analyzer at a heating rate of 10 °C/min under nitrogen atmosphere. The distribution of Fe_3_O_4_ on the surface of rGO was observed by a transmission electron microscopy (TEM, JEOL JEM-2100F, Tokyo, Japan) at an accelerating voltage of 200 kV. The electromagnetic parameters were collected using a vector network analyzer (Agilent HP8753D, Santa Clara, CA, USA). All the samples (35% nanocomposites and 65% wax) were pressed to toroidal-shaped rings (D_inner_: 3 mm, D_outer_: 7 mm).

## 3. Results and Discussion

The crystalline structures of the obtained nanocomposites are investigated (Figure 1). GO shows a typical diffraction peak at 11.7°, which can be attributed to the introduction of oxygenic functional groups on the honeycomb carbon skeleton of graphene. After hydrothermal treatment for 12 h, a new peak appears at 23.3° and the peak at 11.7° disappears in the spectrum of rGO, implying the reduction of GO to graphene. For Fe_3_O_4_, seven peaks located at 18.3°, 30.1°, 35.5°, 43.1°, 53.5°, 57.0° and 62.6° can be assigned to the reflections of (111), (220), (311), (400), (422), (511) and (440) planes of Fe_3_O_4_ (JCPDS, Card No. 75-0033) with the face-centered cubic structure [21]. The spectrum of rGO-Fe_3_O_4_ exhibits similar patterns with that of Fe_3_O_4_ except for a slightly broad peak at nearly 23.3°, because of the combination of Fe_3_O_4_ nanoparticles and rGO.

The chemical structure of GO, rGO, Fe_3_O_4_, and rGO-Fe_3_O_4_ were studied through FTIR spectra (Figure 2). The characteristic absorption bands located at 1717, 1033, 1578, 1398, and 3398 cm^−1^ are associated with C=O, C–O in the epoxide group, C=C in the aromatic ring, and the deformation and stretching vibration of -OH groups, respectively. The spectrum of rGO is consistent with that of GO except for the intensity of these peaks decrease greatly. It is implied that GO is reduced during the solvothermal process. Two bands appear at 2918 and 2867 cm^−1^, corresponding to -CH_2_ and -CH_3_ groups in the residual polyethylene glycol. For Fe_3_O_4_, a characteristic absorption band appears at 535 cm^−1^, corresponding to Fe–O vibrations of Fe_3_O_4_. rGO–Fe_3_O_4_ also exhibits the same curve as Fe_3_O_4_ and rGO, indicating Fe_3_O_4_ nanoparticles were effectively decorated on rGO nanosheets.

Raman is used to further investigate the structural changes of rGO–Fe_3_O_4_ composites. The results are shown in Figure 3. Except for Fe_3_O_4_, the other nanocomposites present the typical reflections of carbon at 1349 cm^−1^ (D band, the disordered structures of GO) and 1589 cm^−1^ (G band, the in-plane vibrations of *sp^2^* bonded carbon atoms). The I_D_/I_G_ is used to measure the quantity of defects. The I_D_/I_G_ ratio of rGO (0.95) is slightly lower than that of GO (0.98), indicating the reduction of GO. The existence of Fe_3_O_4_ nanoparticles may disorder the structure of graphene, so the I_D_/I_G_ ratio of rGO-Fe_3_O_4_ increases up to 1.11. The typical peaks of Fe_3_O_4_ centered around 215, 276, 388, 481, and 585 cm^−1^ appear in Fe_3_O_4_ and rGO–Fe_3_O_4_ spectra, which is assigned to the A_g1_(1), E_g2_ + E_g3_, E_g4_, A_1g_(2), and E_g5_ modes, respectively [22].

The thermal stability of these samples are also investigated (Figure 4). The weight losses of GO are ~10 wt.% (at around 100 °C) and ~40 wt.% (at around 300 °C), indicating the removal of H_2_O and the pyrolysis of the oxygenic functional groups, respectively. The total weight loss of rGO is nearly 30 wt.% at around 400 °C, resulting from the decomposition of residual oxygenic functional groups. The improved thermal stability of rGO demonstrates the reduction of GO. Compared to GO and rGO, Fe_3_O_4_ and rGO–Fe_3_O_4_ exhibit better thermal stabilities, and the weight losses are 6 wt.% and 9 wt.% at 800 °C, suggesting the mass ratio of Fe_3_O_4_ is much larger than that of graphene nanosheets in rGO-Fe_3_O_4_ composites. It is concluded that the obtained rGO-Fe_3_O_4_ nanocomposites will possess an excellent EMW absorption at low thickness because the magnetic Fe_3_O_4_ nanoparticles are preponderant in mass.

Typical transmission electron microscope (TME) of rGO–Fe_3_O_4_ are presents in Figure 5. Fe_3_O_4_ nanoparticles with spherical structures are dispersed on the rGO nanosheets (Figure 5a). The average diameter of Fe_3_O_4_ particles is 11.3 ± 1.8 nm based on the size distribution analysis with 100 arbitrarily selected nanoparticles (Figure 5b). In the hydrothermal process, Fe^3+^ ions are anchored by the rGO nanosheets and grow into Fe_3_O_4_ nanoparticles. The rGO nanosheets play a confinement function to prevent the Fe_3_O_4_ nanoparticles from detaching and aggregating. From the high-resolution transmission electron microscopy (HRTEM) image in Figure 5c, the interplanar spacing (0.253 nm) is assigned to Fe_3_O_4_ (311) plane [23]. The selected area electron diffraction (SAED) in the bottom-left inset of Figure 5c implies the face-centered cubic (fcc) structure of polycrystalline Fe_3_O_4_ nanoparticles. Therefore, the TEM results further confirm the formation of Fe_3_O_4_ nanoparticles on rGO.

As shown in Figure 6, the complex permittivity *ε*_r_ = *ε*′-*jε*″ and complex permeability *μ*_r_ = *μ*′-*jμ*″ of Fe_3_O_4_ and rGO–Fe_3_O_4_ were measured. The values of *ε*′ and *ε*″ for Fe_3_O_4_ and rGO-Fe_3_O_4_ are shown in Figure 6a,b. For rGO–Fe_3_O_4_, the *ε*′ decreased from 14.11 to 11.48, and *ε*″ values increased from 3.13 to 4.67 (in 2–18 GHz). It is obvious that the *ε*′ and *ε*″ for rGO–Fe_3_O_4_ are much larger than that of Fe_3_O_4_. Higher values of *ε*′ are attributed to the existence of conductive rGO nanosheets. The higher *ε*″ values are ascribed to the multiple dielectric loss behaviors of rGO–Fe_3_O_4_. Firstly, the high electric conductivity of rGO-Fe_3_O_4_ is facilitated to increase the dielectric loss. Besides, the Fe_3_O_4_ nanoparticles loaded on rGO and the residual -OH and -COOH groups of rGO can serve as dipoles to enhance the dipole polarization. Furthermore, the multiple interfaces between rGO and Fe_3_O_4_ nanoparticles can enhance the interfacial polarization. The above factors are beneficial for the enhancement of dielectric loss behaviors of rGO–Fe_3_O_4_.

The Debye dipolar relaxation is regarded as a key reason to explain EMW absorption mechanism of materials [24]. The *ε*_r_ can be described as Equation (1) [25]:(1)$$\varepsilon_{r}=\varepsilon_{\infty }+\frac{\varepsilon_{s}-\varepsilon_{\infty }}{1+j2\pi f\tau }=\varepsilon^{\prime }-j\varepsilon^{\prime \prime }$$
where $\varepsilon_{s}$ and $\varepsilon_{\infty }$ are the static and optical permittivity. τ is the relaxation time. Based on the Equation (1), *ε*′ and *ε*″ are expressed as Equations (2) and (3):(2)$$\varepsilon^{\prime }=\varepsilon_{\infty }+\frac{\varepsilon_{s}-\varepsilon_{\infty }}{1+{\left(2\pi f\right)}^{2}\tau^{2}}$$
(3)$$\varepsilon^{\prime \prime }=\frac{2\pi f\tau \left(\varepsilon_{s}-\varepsilon_{\infty }\right)}{1+{\left(2\pi f\right)}^{2}\tau^{2}}$$

The relation of *ε*′ and *ε*″ is expressed as Equations (4) and (5):(4)$${\left(\varepsilon^{\prime }-\frac{\varepsilon_{s}+\varepsilon_{\infty }}{2}\right)}^{2}+{\left(\varepsilon^{\prime \prime }\right)}^{2}={\left(\frac{\varepsilon_{s}-\varepsilon_{\infty }}{2}\right)}^{2}$$
(5)$$\varepsilon^{\prime }=\frac{\varepsilon^{\prime \prime }}{2\pi f\tau }+\varepsilon_{\infty }$$

From Equation (5), the *ε*″ *versus ε*′ curve is a semicircle. Every semicircle is considered as single Debye relaxation process. The *ε*″-*ε*′ plots for Fe_3_O_4_ and rGO–Fe_3_O_4_ are presented in Figure 6c. The curve of Fe_3_O_4_ is irregular, while the curve of rGO–Fe_3_O_4_ exhibits four semicircles, and every semicircle is corresponding to a Debye dipolar relaxation. It suggests that the existence of multiple dielectric relaxations in the electromagnetic wave absorption processes. Besides, the Cole–Cole semicircles are distorted, which implys there are other processes (conductive loss, Maxwell–Wagner relaxation, and dipolar polarization) [26].

The *μ*′ and *μ*″ of Fe_3_O_4_ and rGO–Fe_3_O_4_ are presented in Figure 6d,e. The *μ*′ values of Fe_3_O_4_ exhibit a decreasing trend from 1.28 to 0.88 at 2–8 GHz and remain constant with slight fluctuations over 8–18 GHz. Compared with Fe_3_O_4_, rGO–Fe_3_O_4_ have a similar tendency in *μ*′, which is attributed to a mass of magnetic Fe_3_O_4_ particles on non-magnetic rGO. The values of *μ*″ for Fe_3_O_4_ and rGO–Fe_3_O_4_ sharply decrease with the increasing frequency between 2 GHz and 8 GHz. The peak is mainly ascribed to the natural resonance derived from Fe_3_O_4_ nanoparticles [27]. The resonance frequencies of Fe_3_O_4_ nanoparticles are usually less than 2 GHz [7]. The obtained nanocomposites with nanometer-scale sizes will possess enhanced anisotropy energies and their resonance frequencies can shifts to higher frequencies due to the small size effect [28].

The *μ*″ of Fe_3_O_4_ is approximately constant at 8–18 GHz, while that of rGO–Fe_3_O_4_ becomes relatively stabilized with some fluctuations and increases gradually from 14 GHz to 18 GHz. Eddy current loss is a key factor for magnetic loss. It is verified as follows Equation (6):(6)$$u^{\prime \prime }=\frac{2\pi u_{0}{\left(u^{\prime }\right)}^{2}\sigma d^{2}f}{3}$$
where *μ_0_* is vacuum permeability, *σ* is conductivity, *d* is the thickness. If the eddy current loss is the main factor, the *C_0_* (*C_0_* = 2π*μ_0_σd^2^*/*3*) is independent of frequency. The *C_0_* curve is shown in Figure 6f. *C_0_* for Fe_3_O_4_ sharply decrease at from 2 GHz to 8 GHz and remain constant at 8–18 GHz, indicating that eddy current loss is the key magnetic loss at X and Ku bands. The *C_0_* curve of rGO-Fe_3_O_4_ is similar to that of Fe_3_O_4_ except for the increase from 14 GHz to 18 GHz, implying that there is some other mechanism for rGO–Fe_3_O_4_. It is ascribed to the synergistic effect of rGO and Fe_3_O_4_ nanoparticles, which is similar with Shu’s work [29].

The *tanδ_ε_* and *tanδ_µ_* of Fe_3_O_4_ and rGO-Fe_3_O_4_ are calculated to describe the electromagnetic loss abilities (Figure 7). It is obviously that the plots of *tanδ_ε_* and *tanδ_µ_* have similar trends with those of *ε*″ and *μ*″. The values of *tanδ_ε_* for rGO–Fe_3_O_4_ are much higher than that of Fe_3_O_4_ especially at the Ku band because of rGO nanosheets. Besides, the *tanδ_μ_* value for rGO–Fe_3_O_4_ is smaller than *tanδ_ε_*, implying the dielectric loss is preponderant in the electromagnetic wave absorption, as reported in other literature [30]. The enhancements of *tanδ_ε_* and *tanδ_μ_* at high frequencies are favorable for improving EMW absorbing properties in the high frequency regions.

The reflection loss (RL) curves of Fe_3_O_4_ and rGO-Fe_3_O_4_ at varied thickness are presented in Figure 8. The minimum RL of bare Fe_3_O_4_ is −7.7 dB at 5.2 GHz with 6.5 mm. The EMW absorption performance of rGO nanosheets is also poor due to the absence of the magnetic component [31]. rGO–Fe_3_O_4_ displays the enhanced EMW absorption performance. The minimum RL are −34.4 dB at 1.6 mm and −37.5 dB at 6.5 mm. The corresponding effective bandwidths (≤−10 dB) are 3.8 GHz and 1.9 GHz. The EMW absorption properties of rGO–Fe_3_O_4_ composites in high frequency and low frequency are higher than other similar rGO–Fe_3_O_4_ materials, which are reported in other works [32,33]. Besides, the maximum absorption band moves towards the low frequency when absorber thickness increases.

The relationship between reflection loss (RL) and the frequency are calculated by the Equations (7) and (8):(7)$$RL=20\log\left|\frac{Z_{in}-Z_{0}}{Z_{in}+Z_{0}}\right|$$
(8)$$Z_{in}=Z_{0}\sqrt{\frac{{µ}_{r}}{\varepsilon_{r}}}\;tanh\left[j\left(\frac{2\pi fd}{c}\right)\sqrt{{µ}_{r}\varepsilon_{r}}\right]$$
where *c* is the speed of light, *d* is the thickness of the microwave absorbing material, and *Z* and *Z*_0_ are the input impedance and free space impedance, respectively.

Obviously, rGO-Fe_3_O_4_ nanocomposites exhibit good EMW absorption abilities, which can be attributed to these factors. Firstly, the defects caused by Fe_3_O_4_ nanoparticles and residual hydroxy and carboxyl groups on rGO can act as the dipoles to improve the polarization relaxation. Secondly, the introduction of magnetic Fe_3_O_4_ nanoparticles improves the magnetic loss abilities according to natural resonance and eddy current effect mechanism. Finally, the abundant heterointerfaces between graphene and Fe_3_O_4_ nanoparticles are conducive to the multiple reflections, scatting, and refraction of the incident electromagnetic wave, which results in improved electromagnetic attenuation.

## 4. Conclusions

rGO-Fe_3_O_4_ nanocomposites were synthesized by the hydrothermal method. GO was reduced to rGO and Fe_3_O_4_ nanoparticles and simultaneously loaded on rGO nanosheets. The obtained rGO–Fe_3_O_4_ nanocomposites possessed optimal RL values of −34.4 dB at 1.6mm. The effective absorption bandwidth is 3.8 GHz. Another minimum RL value is −37.5 dB at 6.5 mm with an absorption bandwidth of 1.9 GHz. The conductivity loss, polarization loss, eddy current loss, and natural resonation are major mechanisms in the electromagnetic wave dissipations. It is believed that rGO–Fe_3_O_4_ nanocomposites can be employed as a high-efficiency electromagnetic wave absorbing candidate.

## Figures and Tables

**Figure 1 materials-13-03065-f001:**
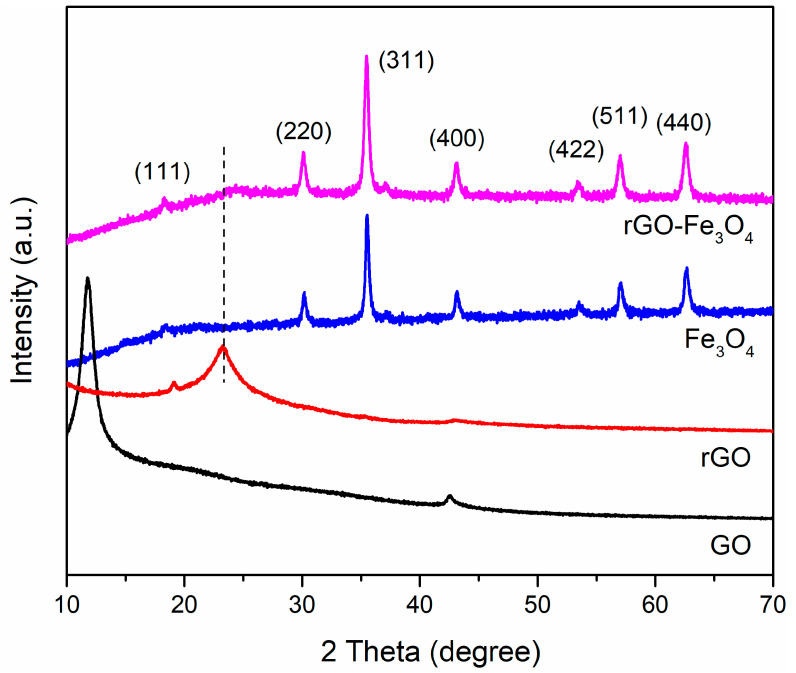
X-ray diffraction patterns of GO, rGO, Fe_3_O_4_, and rGO–Fe_3_O_4_ nanocomposites.

**Figure 2 materials-13-03065-f002:**
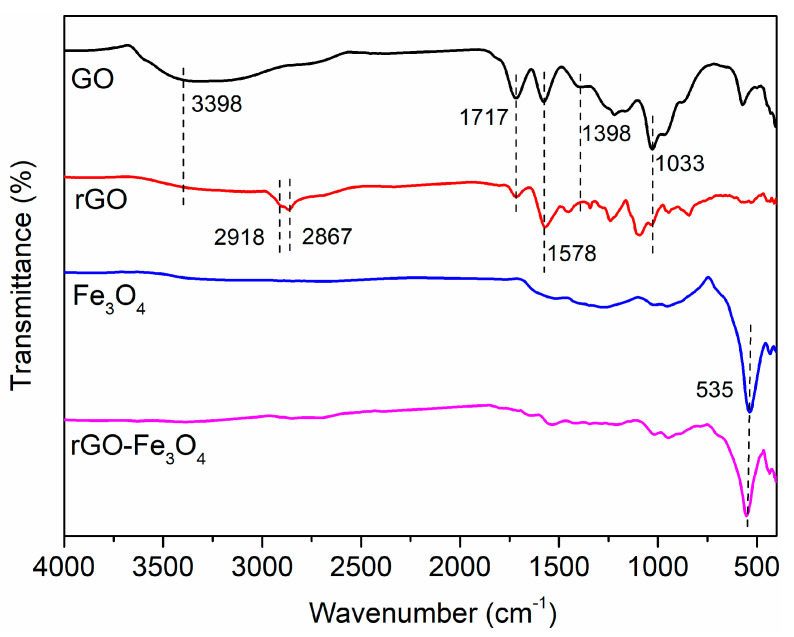
Fourier transform infrared spectra of GO, rGO, Fe_3_O_4_, and rGO–Fe_3_O_4_.

**Figure 3 materials-13-03065-f003:**
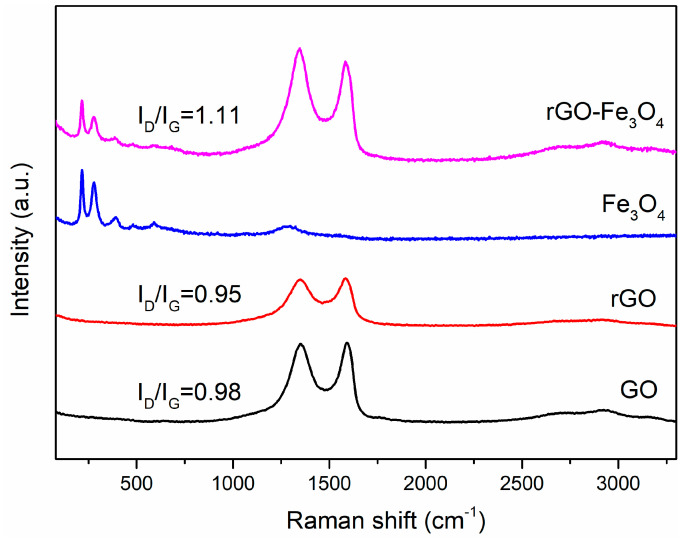
Raman spectra of GO, rGO, Fe_3_O_4_, and rGO–Fe_3_O_4_ nanocomposites.

**Figure 4 materials-13-03065-f004:**
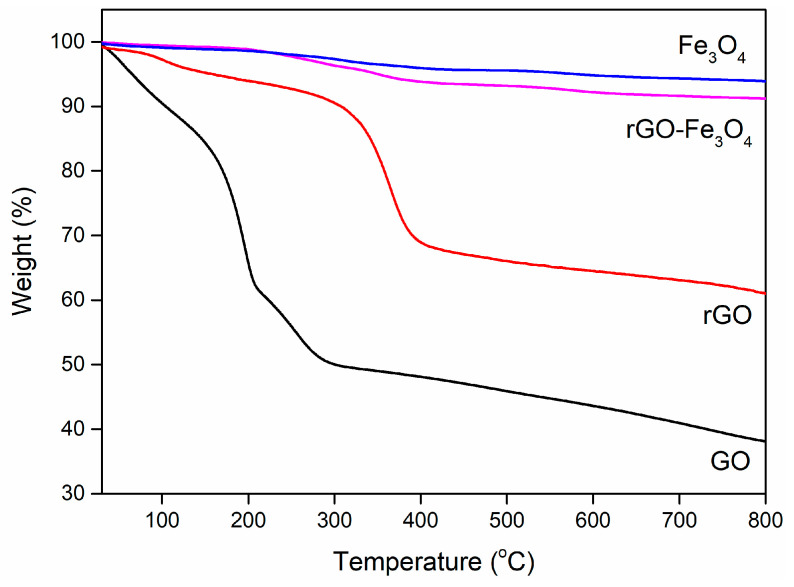
Thermogravimetric analysis curves of GO, rGO, Fe_3_O_4_, and rGO–Fe_3_O_4_ nanocomposites.

**Figure 5 materials-13-03065-f005:**
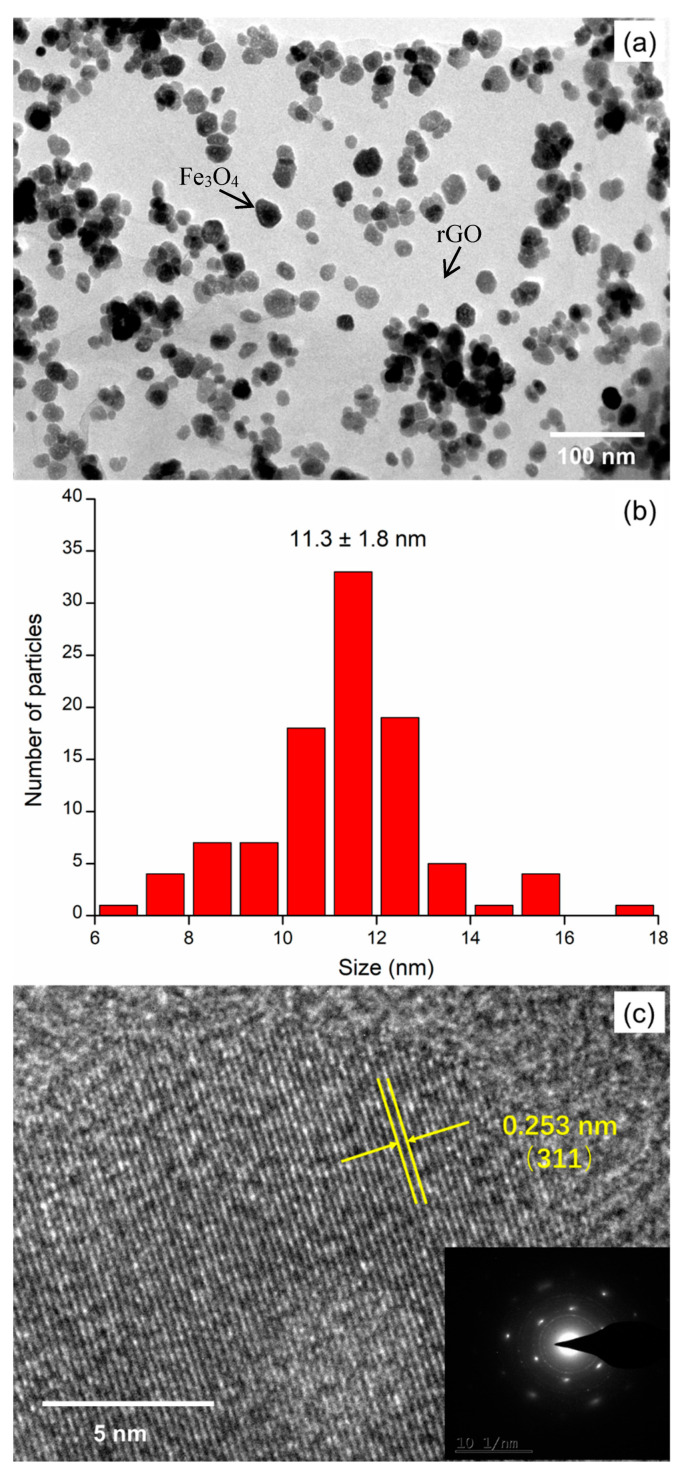
(**a**) Transmission electron microscope image of rGO-Fe_3_O_4_, (**b**) particle size histograms of Fe_3_O_4_ particles, (**c**) high-resolution transmission electron microscopy image of rGO–Fe_3_O_4_, inset shows the selected area electron diffraction pattern.

**Figure 6 materials-13-03065-f006:**
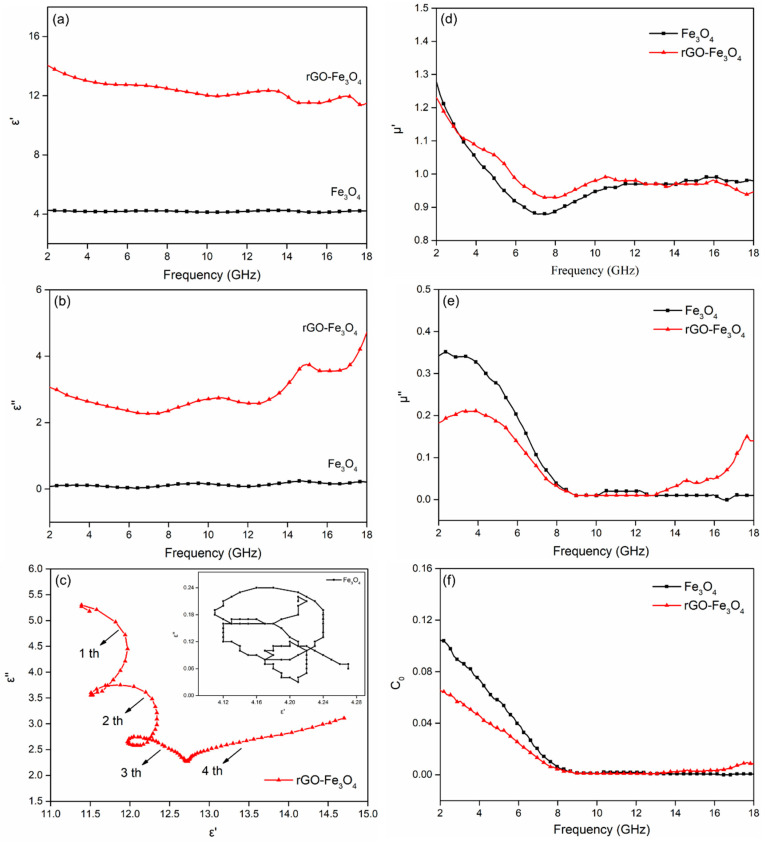
(**a**) Real and (**b**) imaginary parts of complex permittivity, (**c**) Cole–Cole semicircles (*ε*″ *versus ε*′), (**d**) real and (**e**) imaginary parts of complex permeability, and (**f**) C_0_
*versus* frequency for Fe_3_O_4_ and rGO–Fe_3_O_4_.

**Figure 7 materials-13-03065-f007:**
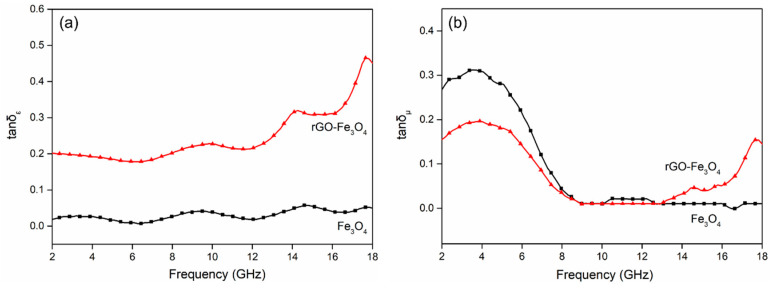
Plots of dielectric loss tangent (*tanδ_ε_*) (**a**) and magnetic loss tangent (*tanδ_μ_*) (**b**) for Fe_3_O_4_ and rGO–Fe_3_O_4_.

**Figure 8 materials-13-03065-f008:**
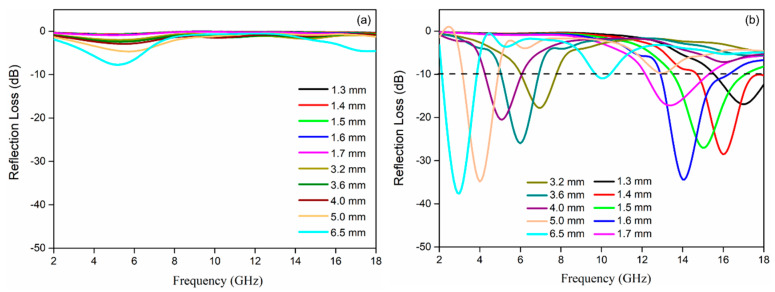
The RL curves of Fe_3_O_4_ (**a**) and rGO–Fe_3_O_4_ (**b**) with various thickness vs. frequency within the range of 2–18 GHz.

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
