# Peer review of "One-Pot Hydrothermal Preparation of Fe_3_O_4_ Decorated Graphene for Microwave Absorption"

_materials, 2020, doi:10.3390/ma13143065_

Round 1

Reviewer 1 Report

The problem considered at work is very important and undertaken for many years by scientists. The article deals with the problem of materials absorbing electromagnetic waves. The problem is current and very significant in the light of the continuous increase in the use of electronic devices. There are science works which try to select and test materials for microwave absorption. The authors place their hope in carbon-based materials due to their low density, thermostability and corrosion resistance. As a solution to the problem of poor graphene impedance, the authors propose a method of producing Fe3O4-rGO nanocomposites

The presented work aim was  preparingFe3O4 nanoparticles decorated rGO nanocomposites via a simple hydrothermal treatment. Experimental tests, as i.e. thermal tests and chemical analysis (FTIR, Raman) were performed.

The manuscript is interesting. However, there are several points that I would like to address:

- the hypothesis should be clearly written,

- the authors did not mention whether similar solutions (e.g. rGO composites and other nanoparticles?) have already been described in the literature,

- the methodology should be found later in the work,

- in the point 2.3 of the methodology, more details related to the conduct of tests as technical parameters of the tests should be given,

- discussion of the experimental results should be more wide, as now it only contains a description of the results without comparison to other materials and other works,

- the article is poorly prepared in terms of formatting (different formatting styles; captions under the drawings do not comply with the guidelines),

- line 54: “wildly” – please correct,

- line 70: “pweders” – please correct,

- values and units should be written in the same line as, in line 175-176.

Author Response

Reviewer: 1

The problem considered at work is very important and undertaken for many years by scientists. The article deals with the problem of materials absorbing electromagnetic waves. The problem is current and very significant in the light of the continuous increase in the use of electronic devices. There are science works which try to select and test materials for microwave absorption. The authors place their hope in carbon-based materials due to their low density, thermostability and corrosion resistance. As a solution to the problem of poor graphene impedance, the authors propose a method of producing Fe3O4-rGO nanocomposites

The presented work aim was preparingFe3O4 nanoparticles decorated rGO nanocomposites via a simple hydrothermal treatment. Experimental tests, as i.e. thermal tests and chemical analysis (FTIR, Raman) were performed.

The manuscript is interesting. However, there are several points that I would like to address:

- the hypothesis should be clearly written,

Reply: Many thanks for the suggestion. The hypothesis of the EMW absorption mechanism was rewritten in the revised manuscript. (Please see the second paragraph in page 10).

- the authors did not mention whether similar solutions (e.g. rGO composites and other nanoparticles?) have already been described in the literature,

Reply: The similar solutions about rGO composites and other nanoparticles in the literature have been mentioned in the revised manuscript. (Please see the second paragraph in page 2).

- the methodology should be found later in the work, in the point 2.3 of the methodology, more details related to the conduct of tests as technical parameters of the tests should be given,

Reply: Thank you for your kind suggestion. The details of the tests were given in the revised manuscript. (Please see the last paragraph in page 2)

- discussion of the experimental results should be more wide, as now it only contains a description of the results without comparison to other materials and other works,

Reply: The experimental results have been discussed more widely. (Please see the last paragraph in page 9)

- the article is poorly prepared in terms of formatting (different formatting styles; captions under the drawings do not comply with the guidelines),

- line 54: “wildly” – please correct,

- line 70: “pweders” – please correct,

- values and units should be written in the same line as, in line 175-176.

Reply: Thank you for your kind suggestion. We have carefully checked and revised all the mistakes throughout the manuscript.

Reviewer 2 Report

The quality of the manuscript is high, the results are well presented, the idea of the authors is also well proved. I just have a few minor comments on the novetly of this study:

  1. The synthesis graphene-Fe3O4 has already been reported extensively, what is new from this study?
  2. The content of Fe3O4 in the composite is important, it is also not clear if it is Fe3O4 on the surface of rGO. If possible, I suggest the authors to get XPS spectra for the composites, which can settles this problem.

Overall, it is a good study and can be accepted after minor corrections. 

Author Response

Reviewer: 2

The quality of the manuscript is high, the results are well presented, the idea of the authors is also well proved. I just have a few minor comments on the novetly of this study:

1. The synthesis graphene-Fe3O4 has already been reported extensively, what is new from this study?

Reply: Thank you for your kind inquiry. The content of Fe3O4 in the composites is optimized in this work in order to obtain a good impedance matching. The obtained rGO-Fe3O4 nanocomposites exhibit good EMW absorption abilities both at high frequency (-34.4 dB at 14.0 GHz) and low frequency (-37.5 dB at 2.9 GHz), which is different from the other work. Some modifications are added at page 9-10 in the revised manuscript.

2. The content of Fe3O4 in the composite is important, it is also not clear if it is Fe3O4 on the surface of rGO. If possible, I suggest the authors to get XPS spectra for the composites, which can settles this problem.

Reply: Thank you for your kind suggestion. The content of Fe3O4 is about 21.6 wt%, which is measured by XPS (Fig. R1).

Fig. R1 XPS spectra of GO, rGO, Fe3O4 and rGO-Fe3O4.

Reviewer 3 Report

Du et al. report a facile one-pot synthesis of magnetite-decorated rGO toward effective microwave EMI shielding. Hybridization of magnetite and rGO are not novel and has been extensively studied. The difference from other reports is the fact that which one is coated on which nanomaterial. In most cases, rGO is decorated with magnetite (for example: RSC Adv., 2019, 9, 20643 and ACS Appl. Mater. Interfaces 2020, 12, 19, 22088–22098). However, this report shows that magnetite has rGO on the nanoparticle surfaces as evident in Figure 5 for TEM micrographs. Hence, authors should say rGO-decorated magnetite, instead of saying Fe3O4-decorated graphene. However, I don’t think that this inversed morphology is beneficial for EMI shielding [ACS Appl. Mater. Interfaces 2020, 12, 19, 22088–22098]. In introduction, authors are strongly encouraged to state why they coated iron oxide NPs with rGO, not in opposite way (iron oxide on rGO). Also, it’s not so clear if very small pieces of RGOs are coated on Fe3O4 or Fe3O4 NPs are wrapped with rGOs. Based on TEMs, the first is the case. But, authors are supposed to clarify the size of rGOs or at least morphology of it (wrapped or not). Then, suggest the mechanism of the morphology and the size of the rGO. Also, compare EMI shielding performance of this system with iron oxide decorated rGO. I can be favorable for publication of this manuscript only if above issues are addressed properly.

In Figure 6, rGO data should be presented for comparison in the data sets.

There are many typos:

Line 70: “GO pweders” should read as “GO powders”

Line 73: authors use both “hours” and “h” in the manuscript. Be consistent.

Line 77: “Rigatu Dmax 2400” should read as “Rigaku D/Max 2400”.

Figure 6c: ‘1th, 2th, 3th " should read as “1st, 2nd, 3rd”

Author Response

Reviewer: 3

Du et al. report a facile one-pot synthesis of magnetite-decorated rGO toward effective microwave EMI shielding. Hybridization of magnetite and rGO are not novel and has been extensively studied. The difference from other reports is the fact that which one is coated on which nanomaterial. In most cases, rGO is decorated with magnetite (for example: RSC Adv., 2019, 9, 20643 and ACS Appl. Mater. Interfaces 2020, 12, 19, 22088–22098). However, this report shows that magnetite has rGO on the nanoparticle surfaces as evident in Figure 5 for TEM micrographs. Hence, authors should say rGO-decorated magnetite, instead of saying Fe3O4-decorated graphene. However, I don’t think that this inversed morphology is beneficial for EMI shielding [ACS Appl. Mater. Interfaces 2020, 12, 19, 22088–22098]. In introduction, authors are strongly encouraged to state why they coated iron oxide NPs with rGO, not in opposite way (iron oxide on rGO). Also, it’s not so clear if very small pieces of RGOs are coated on Fe3O4 or Fe3O4 NPs are wrapped with rGOs. Based on TEMs, the first is the case. But, authors are supposed to clarify the size of rGOs or at least morphology of it (wrapped or not). Then, suggest the mechanism of the morphology and the size of the rGO. Also, compare EMI shielding performance of this system with iron oxide decorated rGO. I can be favorable for publication of this manuscript only if above issues are addressed properly.

Reply: We measured the average particle size (11.3±1.8 nm) of Fe3O4. The size of rGO (micron level) is much larger than that of Fe3O4 particles. So in this work, magnetite Fe3O4 is uniformly loaded on the surface of rGO as most of the literatures, and this can be clearly observed from Figure 5(a).

In Figure 6, rGO data should be presented for comparison in the data sets.

Reply: Thank you for your considerate suggestion. From the literature (Appl. Phys. Lett., 2012, 101, 153108), the values of μ' and μ'' are nearly zero for nonmagnetic graphene (Fig.R2). The EMW absorption of graphene is very weak due to the poor impedance matching. So we didn’t put rGO data in this work.

Fig. R2 Electromagnetic parameters (complex permittivity and permeability) of graphene (a) and (d) from the literature (Appl. Phys. Lett., 2012, 101, 153108)

There are many typos:

Line 70: “GO pweders” should read as “GO powders”

Line 73: authors use both “hours” and “h” in the manuscript. Be consistent.

Line 77: “Rigatu Dmax 2400” should read as “Rigaku D/Max 2400”.

Figure 6c: ‘1th, 2th, 3th " should read as “1st, 2nd, 3rd”

Reply: Thank you for your kind suggestion. We have carefully checked and revised all the mistakes throughout the manuscript.

Round 2

Reviewer 1 Report

Dear Authors,

Thank you for your corrections in the revised version of manuscript.

I recommend it for publication.

Reviewer 3 Report

Authors have addressed previously raised concerns and I now can recommend for publication of this manuscript in Materials.